**Subject Category:**
Biology (whole organism)

behaviour/cognition/evolution

audience effects, communication, dyadic encounters, greeting calls, vocal production

**Author for correspondence:**
Pawel Fedurek
e-mail: pawel.fedurek@stir.ac.uk

†Joint first authorship.

# Behavioural patterns of vocal greeting production in four primate species

Pawel Fedurek[1,2,†], Christof Neumann[2,5,†], Yaëlle Bouquet[2], Stéphanie Mercier[2], Martina Magris[3], Fredy Quintero[2] and Klaus Zuberbühler[2,4]

[1]Division of Psychology, Faculty of Natural Sciences, University of Stirling, Stirling, UK
[2]Institute of Biology, University of Neuchâtel, Neuchâtel, Switzerland
[3]Department of Biology, University of Padova, Padova, Italy
[4]School of Psychology and Neuroscience, University of St Andrews, St Andrews, UK
[5]Laboratoire de sciences cognitives et de psycholinguistique, Département d'études cognitives, ENS, EHESS, CNRS, PSL University, Paris, France

 PF, 0000-0002-6902-708X; CN, 0000-0002-0236-1219;
FQ, 0000-0001-5278-2671

Social animals have evolved a range of signals to avoid aggressive and facilitate affiliative interactions. Vocal behaviour is especially important in this respect with many species, including various primates, producing acoustically distinct 'greeting calls' when two individuals approach each other. While the ultimate function of greeting calls has been explored in several species, little effort has been made to understand the mechanisms of this behaviour across species. The aim of this study was to explore how differences in individual features (individual dominance rank), dyadic relationships (dominance distance and social bond strength) and audience composition (presence of high-ranking or strongly bonded individuals in proximity), related to vocal greeting production during approaches between two individuals in the philopatric sex of four primate species: female olive baboons (*Papio anubis*), male chimpanzees (*Pan troglodytes*), female sooty mangabeys (*Cercocebus atys*) and female vervet monkeys (*Chlorocebus pygerythrus*). We found that female vervet monkeys did not produce greeting calls, while in the other three species, low-ranking individuals were more likely to call than high-ranking ones. The effects of dyadic dominance relationships differed in species-specific ways, with calling being positively associated with the rank distance between two individuals in baboons and chimpanzees, but negatively in mangabeys. In none of the tested species did we find strong evidence for an effect of dyadic affiliative relationships or audience on call production. These results likely reflect deeper evolutionary layers of species-specific peculiarities in social style. We conclude that a comparative approach to investigate vocal

behaviour has the potential to not only better understand the mechanisms mediating social signal production but also to shed light on their evolutionary trajectories.

# 1. Introduction

Why do animals vocalize prior to interacting with each other? One universal function of vocal behaviour is to signal subsequent behaviour [1,2]. Examples include signalling submissive, affiliative or aggressive attitudes, which in turn influences whether social interactions occur and their form [1]. Many functions of calling in this context are more or less directly related to aggression, a serious cost associated with group-living [3]. When two individuals approach each other, the prospects of a physical interaction and, consequently, aggression increase. Thus, many species produce signals before or during such dyadic 'encounters' to reduce the probability of aggression and/or to facilitate friendly interactions [4,5]. If such signals involve vocalizations, they are usually termed 'greeting calls'—broadly defined as calls produced when approaching, or being approached by, another individual [6,7]. Several more specific functions have been proposed for greeting calls, such as reinforcing social relationships (dominance [4,8]; social bonds [9]), reconciling after a conflict [10,11], facilitating reunions [7] or recruiting social support [12].

Although the ultimate functions of greeting calls have been explored in several species, it remains unclear for most species how different factors mediate greeting calls on the production level. One possibility is that, across primate species, signal production during approaches is characterized by universal patterns. For example, signalling might reflect a specific (e.g. affiliative) social relationship between two individuals. Alternatively, greeting patterns could be species-specific, reflecting the peculiarities of a species' evolved social style. For example, signalling submission might be more relevant in despotic species than in tolerant species [13]. This is an important research aspect since exposing such universals and differences has the potential to shed light on signal evolution more generally [2,14]. Similarities, for example, may reveal evolutionarily conserved signalling behaviour, whereas differences may indicate more recent adaptations to varying selection pressures. While there have been studies looking at factors influencing vocal greeting behaviour in single species to identify its function (e.g. [15]), studies directly comparing behavioural patterns associated with vocal greeting production across species are scarce.

Several functions of greeting calls have been described in different primate species, such as signalling submission in chimpanzees (*Pan troglodytes*) and affiliative intent in chacma baboons (*Papio ursinus*) [8,15]. These different functions might be relevant to the way the production of these calls is linked to specific features of an individual, such as its social status, or relationship status between individuals. Therefore, a comparative approach looking at signalling behaviour in several species has the potential to shed light on the relationship between such features of the caller and call function. An important step in this endeavour is to gain an understanding of how particular features of the caller contribute to greeting production in the first place, and then how this relates to the function of these calls, keeping in mind that species differences and similarities might exist on both levels.

The purpose of this study was to focus on this first step and to investigate specific features of vocal greeting production in several primate species. Specifically, we were interested in how individual features, dyadic relationships and the presence and composition of an audience shaped the production of vocal greeting during approaches in four primate species: olive baboons (*Papio anubis*), chimpanzees (*Pan troglodytes schweinfurthii*), sooty mangabeys (*Cercocebus atys atys*) and vervet monkeys (*Chlorocebus pygerythrus*). All four species live in large multi-male/multi-female groups with a largely terrestrial lifestyle and are forest dwellers, and all produce greeting calls during encounters with conspecifics. Baboons produce low-amplitude grunts when approaching another individual for grooming or infant handling [9,16]. In chimpanzees, 'pant grunts' are sequences of calls with varying frequency and amplitude given prior to interacting with dominant individuals [8,17]. Vervet monkeys also produce low-amplitude grunts when approaching other individuals [12,18,19]. In sooty mangabeys, low-pitched grunts and high-frequency twitters are produced in several social contexts, such as when foraging close to others [20], but also when approaching another individual to, for example, initiate grooming, usually accompanied by embracing or other physical contact. Hence, while the acoustic structure of greeting calls differs between these species [7–9], they are all produced in a social context, usually during approaches that precede potential interactions with physical contact between pairs of individuals.

We examined the vocal greeting behaviour of these species focusing on three different features known to be associated with the production of greeting calls. First, at the individual level, we examined how an

**Table 1.** Key terminology employed in the study.

| term | definition |
| --- | --- |
| encounter | an event during which an individual approaches or is being approached by another individual at close distance (adapted according to each species) |
| greeting | a signal given during an encounter |
| greeting call | vocal signal given during encounters (i.e. grunts for baboons and vervets, pant grunts for chimpanzees, and grunts or twitters for sooty mangabeys) |
| target | an individual that is being approached during an encounter |
| approacher | an individual who approaches during an encounter |
| partner | an individual involved in an encounter with the focal animal |
| social role | general behaviour of an individual during an encounter: an individual can either *approach* or *be approached* by another individual |

individual's position in the social hierarchy influenced the probability of calling during approaching, or being approached by, another individual. Whereas in male chimpanzees, low-ranking individuals are considerably more likely to produce greeting calls than high-ranking ones [8,17], it is unclear what the corresponding patterns are in female olive baboons, sooty mangabeys or vervet monkeys. We also investigated whether calling depended on the social role of an individual during an approach, that is, whether the individual was approaching or was being approached [12].

Second, at the dyadic level, we examined whether the probability of calling during an approach was mediated by the dominance relationship between two individuals. In chimpanzees, for example, it is almost always the lower-ranking individual that calls towards a higher-ranking partner [8]. On the other hand, in chacma baboons, higher-ranking individuals often direct calls towards lower-ranking ones apparently to signal benign intent [16]. In sooty mangabeys and vervet monkeys, however, this aspect of greeting calls has not been investigated yet. We further investigated the effect of social bond strength, a dyadic feature with demonstrated effects on primate vocal behaviour (e.g. [21,22]), on the occurrence of greeting calls. For instance, female chacma baboons produce greeting calls mainly towards individuals with whom they have weak social relationships compared to strongly bonded group members [15]. However, it is unclear whether the same applies to the four investigated species.

Third, at the triadic level, we looked at the role of the audience, such as the presence of high-ranking and affiliated individuals, on greeting behaviour—a topic virtually unexplored in the literature. One notable exception is a study by Laporte *et al.* [6], who showed that female chimpanzees were less likely to produce greeting calls in the presence of the most high-ranking male in the community. However, it is unclear whether the same applies to male–male interactions in this species and whether similar patterns characterize other species. Similarly, little is known as to whether in any primate species, the presence of bonded individuals in the audience affects greeting behaviour.

To summarize, the goal of this study was to investigate correlates of the occurrence of greeting calls during approaches on three levels: individual, dyadic and triadic. It is important to note that while we refer sometimes to the function of greeting calls in particular species, the purpose of this study was not to investigate the ultimate function of greeting calls in the four species but to explore mechanisms underpinning the production of these calls or their immediate correlates.

# 2. Material and methods

## 2.1. Study sites and subjects

We collected data on dyadic encounters (i.e. between a focal animal and another individual, table 1 for definitions of the key terms used in the study) in the four species. We limited our data collection to the philopatric sex, that is, males in chimpanzees and females in the other three species. We did so because it was not always possible to describe dyadic attributes, such as affiliative relationships, in the non-philopatric sex due to frequent migrations. In total, we collected and analysed data on 813 approaches (table 2).

**Table 2.** Overview of the data collected.

| species | social role of focal individual | N encounters | N encounters with vocalization by the focal | mean calling proportion across individuals | N individuals for calling proportion |
|---|---|---|---|---|---|
| baboon | approacher | 140 | 44 | 0.32 | 10 |
| (N = 10 | target | 133 | 12 | 0.08 | 10 |
| individuals) | total | 273 | 56 | 0.19 | 10 |
| chimpanzee | approacher | 94 | 13 | 0.16 | 11 |
| (N = 11 | target | 145 | 36 | 0.33 | 11 |
| individuals) | total | 239 | 49 | 0.26 | 11 |
| mangabey | approacher | 97 | 21 | 0.23 | 18 |
| (N = 18 | target | 143 | 17 | 0.09 | 17 |
| individuals) | total | 240 | 38 | 0.16 | 18 |
| vervet | approacher | 32 | 0 | 0 | 10 |
| (N = 10 | target | 29 | 0 | 0 | 8 |
| individuals) | total | 61 | 0 | 0 | 10 |
| total | approacher | 363 | 78 | 0.19 | 49 |
| (N = 49 | target | 450 | 65 | 0.13 | 46 |
| individuals) | total | 813 | 143 | 0.16 | 49 |

### 2.1.1. Olive baboons

YB collected data on the Kabasinguzi troop at the Kanyawara study site, Kibale National Park, Uganda, from May until December 2015. The troop was fully habituated to human presence [23] and all individuals were individually identified. During the study period, the group included between 39 and 44 individuals [24]. Study subjects were adult (individuals that had already given birth to their first infant; $N = 8$: $\geq 4$–6 years) and subadult (animals reaching menarche that had not yet given birth but had full swellings; $N = 2$: $\geq 3$ years; [25]) females.

### 2.1.2. Chimpanzees

PF collected data on the Sonso community of Budongo Forest, Uganda, also from May to December 2015. The group was also fully habituated to human presence [26]. At the time of the study, the community contained 75 individuals, with a core home range of around 15 km². Study subjects were adult ($N = 9$: $\geq 16$ years) and late adolescent ($N = 2$: $\geq 13$–15 years; [17]) males.

### 2.1.3. Sooty mangabeys

MM collected data on the ATY1 group of Taï National Park, Ivory Coast, from February to July 2014. The study group was well habituated to human observers [27,28]. During the study period, the group size was around 80 individuals. Study subject were adult females ($N = 18$: $\geq 5$ years; [29]).

### 2.1.4. Vervet monkeys

SM collected data on two wild groups of vervet monkeys at the Mawana Game Reserve, Kwa Zulu-Natal, South Africa from July 2014 to March 2015. Both groups were well habituated to human observers. During the study period, group sizes varied from 45 to 56 individuals [30]. Study subjects were adult ($N = 10$: $\geq 5$ years) females.

## 2.2. Data collection

Each day, a randomly chosen focal individual was followed for the whole day by the respective observers. *Encounters* took place when the focal individual approached or was approached by another

individual (hereafter: *partner*) at a distance of 5 m (olive baboons and vervet monkeys), or 10 m (chimpanzees), or 0 m (sooty mangabeys) [6]. The identities of the partner and of other individuals (the *audience*) present within 10 m (olive baboons, sooty mangabeys and vervet monkeys) or 15 m (chimpanzees) of the focal were also recorded. We used different distance-based criteria for encounters and audience to better reflect species differences based on observations during pilot studies. We noted whether or not the focal animal or the partner produced a greeting call and the social role of the focal and the partner during an encounter (i.e. whether they approached or were being approached, table 1).

### 2.2.1. Olive baboons

To assess the strength of affiliative relationships, we used focal animal sampling [31]. In addition, we used instantaneous scan samples at 15 min intervals to record (1) the identity of the nearest individual from the focal animal and (2) the identities of all individuals present within 5 m. We established the dominance hierarchy based on displacements, unidirectional fear barks (i.e. vocalizations given by subordinates towards dominants only [32]) and decided aggressive interactions (i.e. when the outcome of the agonistic interaction was clear, with a winner who displaced or chased another one and a loser who is displaced or chased).

### 2.2.2. Chimpanzees

To establish the strength of social relationships between males, we collected instantaneous scan samples [31] at 15 min intervals to record (1) the identities of individuals present in the focal individual's party (defined as all adult and late adolescent individuals present within 35 m of the focal animal [33]), (2) the identities of adult and late adolescent males present within 5 m of the focal male and (3) the identity of the adult or late adolescent male closest to the focal male. To calculate the dominance hierarchy of the males, all-occurrence data on agonistic interactions such as displacement, physical attack, chase, charge, give ground or submissive crouch (e.g. [34]) were used.

### 2.2.3. Sooty mangabeys

To calculate social bond strength between females, we used data from focal animal sampling [31]. During focal follows we recorded grooming interactions continuously. In addition, we used instantaneous scan samples at 15 min to recorded proximity data, i.e. the identity of the nearest adult female within 5 m around the focal individual and the identities of all adult females within 5 m of the focal animal. Data on all occurrences of decided dyadic conflicts were recorded and subsequently used to calculate the female dominance hierarchy.

### 2.2.4. Vervet monkeys

To assess the strength of affiliative relationships, we used focal animal sampling [31]. During instantaneous samples collected every 15 min, we recorded the identity of the nearest female around the focal animal, the identities of all females present within 5 m of the focal animal and all affiliative interactions (grooming, sitting in contact and mouth to mouth contact) occurring between the focal and another identified female. To calculate the dominance hierarchy, we recorded displacements and decided aggressive interactions (i.e. interactions with a clear winner who displaced or chased another individual).

## 2.3. Data processing

### 2.3.1. Individual features

For each encounter, we extracted the identity, dominance status and role of the focal animal (i.e. approaching versus being approached, table 1) and its partner. Dominance status was estimated with Elo-rating [35,36]. In brief, this method assigns ratings on an interval scale to individuals and these ratings typically correlate highly with ordinal ranks [37]. The calculation process starts with each individual being assigned the same (arbitrary) rating. Subsequently, as each dominance interaction is evaluated progressively, ratings of individuals change: winners of dominance interactions/fights increase in ratings and losers decrease in ratings. The amount of change in ratings is determined by the expectation of the outcome of an interaction prior to that interaction: a highly expected outcome (a high-rated individual wins against a low-rated individual) will lead to small changes in both

individuals. By contrast, a highly unexpected outcome (a low-rated individual wins against a high-rated individual) will lead to relatively larger changes in the ratings of the two individuals.

### 2.3.2. Dyadic features

For dyadic features, we calculated the differences in Elo-ratings between focal animal and partner from the focal animal's perspective [36]. Here, positive values indicated that the focal animal had a higher status than the partner, while negative values indicated the opposite. We estimated social relationship strength for baboons, mangabeys and vervets using a dyadic composite social index (DSI) [38]. Here, large values indicated a strong bond between two individuals regardless of their roles during encounters, and smaller values indicated weak social bonds. To calculate the DSI, we used three behavioural indices: (1) grooming, (2) the identity of the closest individual during instantaneous sample and (3) the identities of all individuals within 5 m. For chimpanzees, we calculated social bond strength on the basis of three different dyadic association measures (simple ratio index, 5 m association index, and nearest neighbour association index, see ref. [21] for details). We first standardized each measure across all dyads to a mean of 0 and a standard deviation of 1. Our composite measure of relationship strength for a given dyad was then calculated as the mean of these three indices for each dyad.

### 2.3.3. Triadic features

Finally, at the triadic level, we described the audience at the beginning of the encounter, from the focal animal's perspective. To this end, we scored whether there was a bonded individual of the focal animal in the audience or not (i.e. at least one of the top three social partners, with which the focal individual had the strongest bond [39]), and whether there was a high-ranking individual in the audience or not (i.e. at least one of the three highest-ranking individuals, as indicated by Elo-ratings).

## 2.4. Statistical analysis

We fitted a generalized linear mixed model with binomial error structure and logit link function to these data [40]. The response variable was whether the focal animal produced a greeting call or not. In order to address our question, we fitted six major predictor variables: (1) Elo-rating of focal individual, (2) the social role of focal individual in an encounter, (3) Elo-rating difference with partner, (4) bond strength with partner, (5) presence of at least one bonded individual in audience and (6) presence of at least one high-status individual in audience. Variables (1) and (2) represent individual features, variables (3) and (4) represent dyadic features and variables (5) and (6) represent triadic features from the focal animal's perspective. Furthermore, we added species as a predictor variable. Since our interest was to differentiate effects that are similar across species from those that differ between species, we fitted the two-way interactions between species and our six main predictors. Finally, we found it likely that species differ with respect to which individual is more likely to call according to their social role. Therefore, we also fitted three-way interactions between the social role of the focal animal and the interactions described so far. In other words, our initial model contained five three-way interactions (e.g. species × role × focal Elo-rating, species × role × Elo-rating difference). We fitted random intercepts for focal animal identity and partner identity. Since this model structure was already quite complex, we restricted random slopes to the following terms, which we considered most crucial: role in focal animal and partner animal identity, bond strength in focal animal identity and Elo-difference in focal identity. Random slopes were fitted without accounting for correlations between slopes and intercepts (table 4).

We transformed the three numerical predictors (Elo-rating, Elo-rating difference and bond strength) where necessary to achieve symmetric distributions and subsequently standardized all variables to mean of 0 and standard deviation of 1. We applied this process for each predictor separately for each species, i.e. we transformed and standardized within species.

We compared the full model to a null model, which contained the same random effects structure as the full model and species as the only fixed effect, with a likelihood ratio test (LRT, [41]). If this full model revealed significance, we removed non-significant interaction terms using LRTs until we reached a model with interpretable terms, i.e. with significant interaction terms and/or main effects (either significant or non-significant) [42–44]. For interpretation, we used this reduced model. We also present graphical results of the full model (electronic supplementary material, figure S1) [43].

All statistical analyses were conducted using R, v. 3.4.3 and the lme4 package, v. 1.1–17 [45,46].

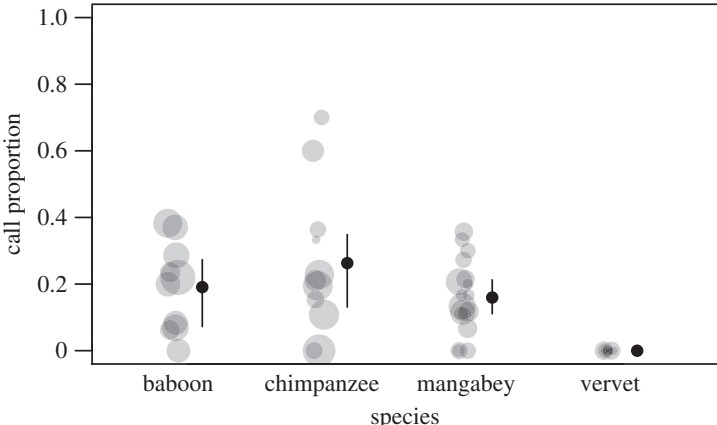

**Figure 1.** Mean call proportions of individuals per species. Grey circles represent individuals and circle size is proportional to the number of encounters observed for each individual. Black circles and lines represent mean and quartiles.

## 3. Results

Table 2 shows the total number of encounters, the number of encounters in which a greeting call was produced and the number of individuals recorded during an encounter depending on the role of the focal animal and the species. Overall, subjects produced greeting calls in about 16% of encounters, but this varied considerably between species, from 0% for vervets to 26% for chimpanzees (figure 1 and table 2). Because in this study vervet females never produced grunts towards other females, we excluded vervets from the remaining analyses.

The full model comprising the six factors of interest and their interactions that may be related to calling during an encounter was significantly different from the null model (LRT: $\chi^2_{33} = 75.86$, $p < 0.001$). The results of the full model and the final model from which all non-significant interaction terms were removed are in tables 3 and 4 (A graphical presentation of the results of the full model is shown in electronic supplementary material, figure S1).

We found that individuals with higher Elo-ratings were significantly less likely to call than individuals with lower Elo-ratings (LRT: $\chi^2_1 = 8.73$, $p = 0.003$; figure 2). This effect was largely independent of species and role, i.e. all interactions of individual Elo-rating with species and role were non-significant.

We found species differences regarding the social role of individuals, i.e. approacher versus target (LRT: $\chi^2_2 = 17.76$, $p < 0.001$). In baboons and mangabeys, approaching individuals were more likely to call than targets, whereas in chimpanzees, we found the opposite (figure 3).

At the dyadic level, Elo-rating differences also affected calling probability, but in species-specific ways (LRT: $\chi^2_2 = 15.76$, $p < 0.001$; figure 4). In chimpanzees and baboons, the smaller the rating difference the higher the probability to call (negative rating differences imply that the focal is lower-ranking than the target, positive rating differences imply the opposite), although the effect was much less pronounced in baboons. In mangabeys, we found the opposite pattern (figure 4).

In all three species, there was no statistically significant relationship between calling probability and the strength of the affiliative relationship with the partner (figure 5, LRT: $\chi^2_1 = 2.34$, $p = 0.126$).

Finally, at the triadic level, we found that audience composition did not significantly affect calling probability of our subjects (table 3; strongly bonded individual in audience, LRT: $\chi^2_1 = 0.67$, $p = 0.413$; high-ranking individual in the audience, LRT: $\chi^2_1 = 0.48$, $p = 0.489$).

## 4. Discussion

Despite the fact that, during approaches, female vervet monkeys grunt sometimes towards adult males as shown in a previous study [12], we did not observe greeting calls *between* females—the philopatric sex in this species that our study focused on. This suggests that the occurrence of greeting calls between vervet females is rare or perhaps even completely absent. Consequently, we excluded vervet monkeys from the analyses. In the other three species (olive baboons, chimpanzees and sooty mangabeys), low-ranking individuals were more likely to call during encounters than high-ranking ones. By contrast, the relationship between calling probability and dominance distance between two individuals differed

**Table 3.** Results of the model investigating individual, dyadic and triadic features related to calling probability. The table contains parameter estimates ± s.e. for the full model and for the final model, from which non-significant interaction terms were removed. For categorical predictors (species and role), the tested levels are indicated in parentheses.

| | full model | final model |
|---|---|---|
| intercept | −1.03 ± 0.39 | −1.11 ± 0.34 |
| species (chimpanzee) | −1.99 ± 0.98 | −1.60 ± 0.65 |
| species (mangabey) | −0.32 ± 0.56 | −0.11 ± 0.48 |
| role (target) | −1.81 ± 0.61 | −1.48 ± 0.43 |
| Elo-difference | −0.17 ± 0.40 | −0.21 ± 0.30 |
| bond strength | 0.04 ± 0.34 | 0.29 ± 0.18 |
| strongly bonded in audience (yes) | −0.39 ± 0.51 | −0.26 ± 0.30 |
| high-rank in audience (yes) | −0.55 ± 0.65 | −0.25 ± 0.35 |
| Elo-rating | −0.87 ± 0.42 | −0.69 ± 0.22 |
| species (chimpanzee) : role (target) | 3.42 ± 1.16 | 2.88 ± 0.69 |
| species (mangabey) : role (target) | 0.79 ± 0.84 | 0.40 ± 0.65 |
| species (chimpanzee) : Elo-difference | −2.32 ± 1.04 | −1.63 ± 0.53 |
| species (mangabey) : Elo-difference | −0.10 ± 0.60 | 0.47 ± 0.38 |
| species (chimpanzee) : bond strength | −0.13 ± 0.71 | |
| species (mangabey) : bond strength | 0.16 ± 0.51 | |
| species (chimpanzee) : strongly bonded in audience (yes) | 0.70 ± 1.18 | |
| species (mangabey) : strongly bonded in audience (yes) | 0.82 ± 1.13 | |
| species (chimpanzee) : high-rank in audience (yes) | 0.29 ± 1.36 | |
| species (mangabey) : high-rank in audience (yes) | 1.32 ± 1.28 | |
| species (chimpanzee) : Elo-rating | 0.49 ± 0.75 | |
| species (mangabey) : Elo-rating | 0.72 ± 0.66 | |
| role (target) : Elo-difference | 0.58 ± 0.58 | |
| role (target) : bond strength | −0.19 ± 0.42 | |
| role (target) : strongly bonded in audience (yes) | 0.14 ± 0.99 | |
| role (target) : high-rank in audience (yes) | 0.11 ± 1.16 | |
| role (target) : Elo-rating | −0.98 ± 0.76 | |
| species (chimpanzee) : role (target) : Elo-difference | −0.07 ± 1.20 | |
| species (mangabey) : role (target) : Elo-difference | 0.79 ± 0.96 | |
| species (chimpanzee) : role (target) : bond strength | 0.90 ± 0.76 | |
| species (mangabey) : role (target) : bond strength | 0.47 ± 0.65 | |
| species (chimpanzee) : role (target) : strongly bonded in audience (yes) | −0.58 ± 1.59 | |
| species (mangabey) : role (target) : strongly bonded in audience (yes) | −1.21 ± 1.84 | |
| species (chimpanzee) : role (target) : high-rank in audience (yes) | 0.67 ± 1.85 | |
| species (mangabey) : role (target) : high-rank in audience (yes) | −1.67 ± 2.00 | |
| species (chimpanzee) : role (target) : Elo-rating | 1.20 ± 1.03 | |
| species (mangabey) : role (target) : Elo-rating | −0.30 ± 1.12 | |

between species, with baboons and chimpanzees calling towards higher-ranking partners and mangabeys calling predominantly towards lower-ranking individuals. Similarly, we identified between-species differences in terms of the social role of the focal in an encounter and calling probability, with olive baboons and sooty mangabeys calling usually when approaching, while chimpanzees calling when being approached by another individual. By contrast, the affiliative

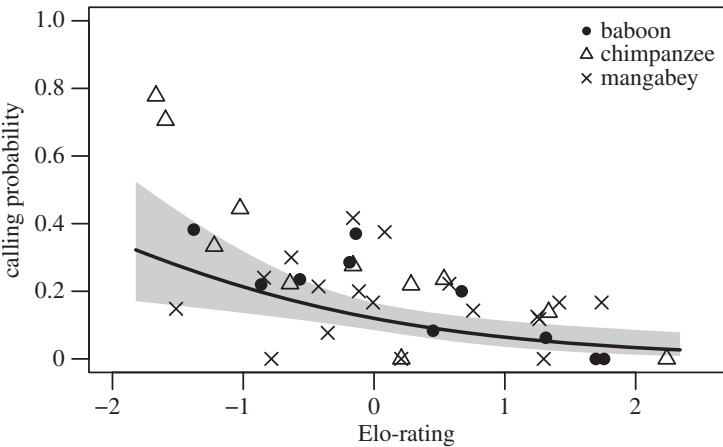

**Figure 2.** The relationship between calling probability and the Elo-rating score in olive baboons, chimpanzees and sooty mangabeys. Each symbol represents an individual, showing its Elo-rating and the proportion of encounters in which it vocalized. The line and shaded area represent the fitted model and the 95% confidence area.

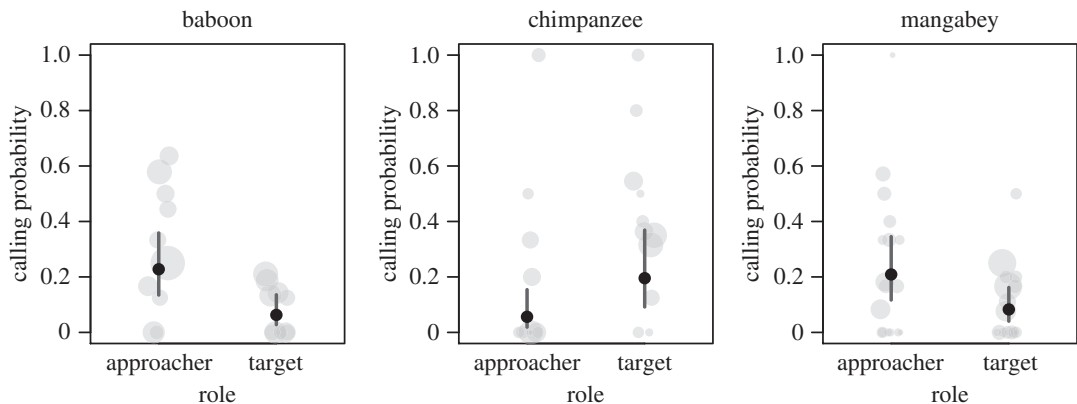

**Figure 3.** The relationship between calling probability and the social role of the focal animal in an encounter in olive baboons, chimpanzees and sooty mangabeys. Shown are model estimates (black circles) and 95% confidence intervals. Raw data are presented as grey circles where each circle represents one individual (circle size is proportional to sample size).

**Table 4.** Random effects structure of our GLMM. Shown are the standard deviations of random intercepts and random slopes.

| grouping | variable | full model | final model | null model |
|---|---|---|---|---|
| focal | intercept | 0.48 | 0.41 | 0.37 |
| | role | 0.33 | 0.35 | 0.92 |
| | bond strength | 0.64 | 0.60 | 0.87 |
| | Elo-difference | 0.00 | 0.20 | 1.17 |
| partner | intercept | 0.50 | 0.47 | 0.70 |
| | role | 0.00 | 0.00 | 0.54 |

relationship between the two individuals, as well as the presence of both high-ranking and affiliated individuals in the audience, appeared to have no considerable effects on greeting call production.

Across the three species, low-ranking individuals were more likely to produce greeting calls than high-ranking ones, largely independent of role. One reason for this finding could be that potential consequences, such as receiving aggression for not producing a greeting call, are higher for low-ranking individuals than for high-ranking ones. Therefore, producing greeting calls, regardless of their specific functions in different species (e.g. signalling submission or benign attitude), might be a strategy used mostly by low-ranking individuals to reduce the likelihood of receiving aggression when approaching others.

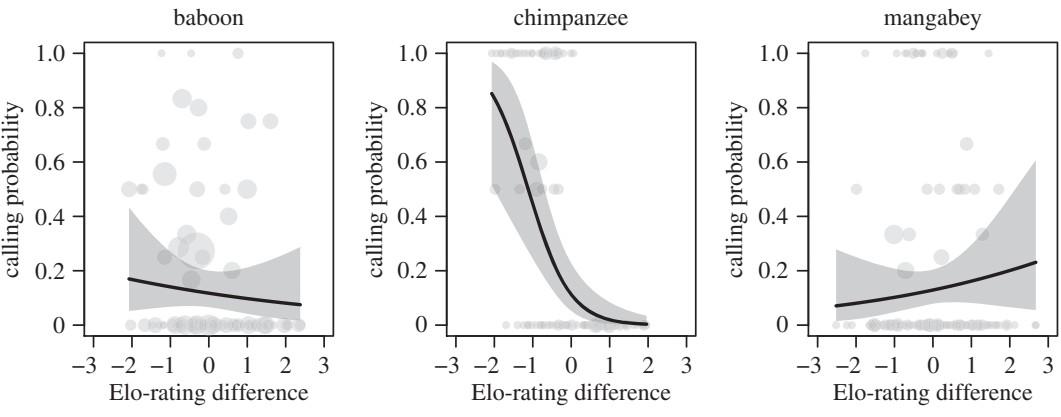

**Figure 4.** The relationship between calling probability and Elo-rating differences between the focal animal and the partner in olive baboons, chimpanzees and sooty mangabeys. The line and shaded area represent the fitted model and the 95% confidence area. Raw data are presented as grey circles where each circle represents one dyad (circle size is proportional to sample size).

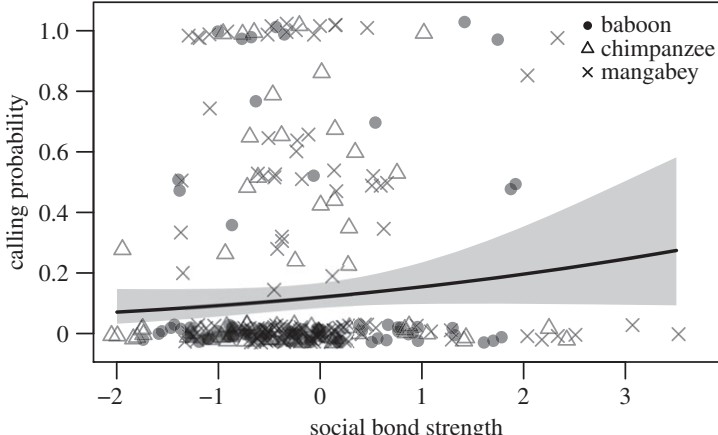

**Figure 5.** The relationship between calling probability and bond strength in olive baboons, chimpanzees and sooty mangabeys. Each symbol represents a dyad of individuals. The line and shaded area represent the fitted model and the 95% confidence area.

The relationship between greeting call production and dominance distance between the two individuals, however, differed substantially between species. In chimpanzees, for instance, if a greeting call was produced, it was the lower-ranking individual who called towards the higher-ranking one, which is consistent with a large body of literature on chimpanzee pant-grunting behaviour [8,34,47]. In olive baboons, similar to chimpanzee males, lower-ranking females produced these calls more often towards higher-ranking partners. However, in contrast to chimpanzees, this pattern was considerably less pronounced since, similar to sooty mangabeys, higher-status individuals also regularly called towards lower-ranking ones. In mangabeys, on the other hand, a higher-ranking individual was more likely to produce these calls towards a lower-ranking partner than vice versa.

The reason for these differences might be attributed to differences in terms of the specific functions of greeting calls suggested for the three species. In chimpanzees, for example, signalling submission to avoid aggression from a higher-ranking partner appears to be the main function of this behaviour [6], which may explain why it is predominantly the lower rather than the higher-ranking individual that exhibits it. In the other two species, greeting calls are unlikely to be involved in agonistic or socio-negative signalling, but might rather serve to signal benign intent, which could explain why higher-ranking individuals often produce greeting signals towards lower ones. In chacma baboons, for example, grunts seem to signal friendly intent towards the approached individual, most likely to reduce fear in the partner and to facilitate friendly interactions [16,38]. Indeed, in chacma baboons, higher-ranking individuals produce greeting calls towards lower-ranking ones more often than vice versa [9].

Comparatively, little is known about the nature of sooty mangabey greeting calls. Since, when produced during encounters, these calls are associated with affiliative interactions such as embracing and grooming [20], mangabey greeting calls, as in baboons, seem to reflect friendly intent rather than

agonistic relationships. Our results also suggest that, in this particular context, chimpanzee greeting calls are considerably less flexible compared to the two monkey species because greeting calls are almost exclusively given towards a higher-ranking partner.

In contrast to dominance differences, the strength of social bonds between individuals appeared to have no pronounced effect on calling, suggesting that greeting calls neither imply nor require strong social bonds in either species. This result contrasts with more intimate forms of greeting involving physical contact, such as in Tonkean macaques (Macaca tonkeana), Guinea baboons (Papio papio) or spotted hyenas (Crocuta crocuta) [48–50]. Interestingly, a study on chacma baboons showed that females were less likely to produce calls when interacting with preferred social partners than with individuals with weak social relationships, probably because the outcome of an encounter with an unaffiliated individual is less predictable [15]. It appears therefore that vocal greetings in our study species have little to do with long-term social bonds. It is important to stress, however, that the purpose of this study was not to explore directly the function of greeting calls in the four species but rather to examine the proximate mechanisms underpinning their production. It is also worth noting that our intention was to gather comparable data to obtain measures of dominance relationships and social bonds. Hence, we relied on established and partly species-specific observation and analysis methods that aimed at maximizing social–ecologic validity with regards to a given species. Whether these differences affect our conclusions remains unknown and future work should show whether such analytical variation influences study conclusions in a meaningful way (e.g. via simulations, [37]).

Across all three species, we found no significant relationship between the presence of a high-ranking or well-affiliated individual and greeting call production. This suggests that the effect of an audience on this behaviour is subtle at best, with the dyadic dominance relationships between individuals during an encounter having a much stronger effect. A previous study on chimpanzees showed that adult females were less likely to give pant grunts to a male when the top-ranking male was nearby [6]. However, in their analyses, Laporte et al. [6] did not consider simpler explanations for call production, such as dyadic features between female and male partner. Our more comprehensive approach suggests that, when compared directly, dyadic features such as dominance difference between two individuals can have a substantially higher impact on greeting call production than triadic features such as the presence of a high-status or strongly bonded audience. On a more general note, simpler explanations for a given phenomenon should be considered prior to exploring potentially more complex effects (e.g. audience effects) on signal production (e.g. greeting calls) [51,52]. In addition, in this study, we looked only at male–male interactions, which may differ from male–female interactions in terms of audience effects on greeting behaviour in chimpanzees. It appears that among male chimpanzees, as in female baboons and sooty mangabeys, the presence of bystanders does not constitute a major selection pressure shaping vocal greeting behaviour.

Finally, we found between-species differences in terms of the social role of an individual in an encounter. More specifically, chimpanzees that were being approached were more likely to produce a greeting call than when they were approaching others, whereas in baboons and mangabeys the converse was the case. Again, one way to explain these differences is by considering the specific function attributed to greeting calls in particular species. If reassuring the partner about the friendly intent of the caller is an important function of these calls, as suggested to be the case in baboons [15], we would expect that the approaching individual would call more than the target. This is, however, not necessarily the case if these calls reflect the caller's submission, as in chimpanzees [8]. Here, we would expect that an individual would often call also when being approached, especially unexpectedly, by a higher-ranking individual, most likely to avoid receiving aggression from him. Our data seem to support this view.

We did not include the context of calling in our analysis (e.g. aggressive or affiliative), since actual interactions often follow rather than precede greetings and frequently interactions do not even occur after approaches [53]. Thus, including such pre-defined contexts would not be feasible for our analyses and would invalidate the temporal sequence of events. In addition, it would be challenging to assign a context to those encounters that were not followed by an interaction. Also, examining the context of calls would be more suitable for examining the function of greeting calls. Again, however, exploring the function of these calls was beyond the scope of this study—future studies should compare greeting behaviour across species from the functional perspective.

Although more species need to be considered to infer evolutionary trajectories of greeting calls, our results are consistent with the view that some features of this behaviour may be evolutionarily ancient. For example, in all three species, low-ranking individuals were more likely to produce greeting calls, irrespective of all other characteristics we investigated. It is thus possible that, in the evolutionary past, selection pressures on displaying this behaviour were stronger for individuals with a low social

standing, and that this behaviour initially evolved, for example, to avoid being a target of aggression. The differences found in this study, on the other hand, suggest that in the course of evolution, this behaviour differentiated on a functional level, to effectively fulfil socially different roles in different species. Ultimately, these differences are likely driven by different patterns of sociality characterizing different species, such as the level of intra-group competition. Future studies should incorporate more species to explore in more detail factors shaping vocal greetings. Such analyses ideally should also include non-primate mammal and avian species where such signals were recorded, furthering our understanding of the evolutionary trajectories of these signals.

Data accessibility. Data available from the Dryad Digital Repository: https://doi.org/10.5061/dryad.jj174p1. Additional information is available in the electronic supplementary material.

Ethics. The study was approved by the University of Neuchatel Ethics Committee, and permission to conduct the study was granted by the Uganda Wildlife Authority (ref: ADM 154/212/03 and COD/96/02), the Uganda National Council for Science and Technology, by the Ministère de la Recherche Scientifique and the Ministère de l'Agriculture et des Ressources Animales in Côte d'Ivoire, as well as by the relevant local authority in South Africa, Ezemvelo KZN Wildlife.

Authors' contributions. P.F., C.N. and K.Z.: study design; P.F., Y.B., S.M. and M.M.: data collection; C.N.: statistical analysis; P.F., C.N., Y.B., S.M., F.Q. and K.Z.: interpretation and drafting the article; K.Z.: provision of necessary tools and resources. All authors read and approved the final manuscript.

Competing interests. The authors declare no competing interests.

Funding. The study was funded by Swiss National Science Foundation (310030_143359) and European Research Council (PRILANG 283871) project grants awarded to K.Z.

Acknowledgement. We are grateful to the management and staff of the Budongo Conservation Field Station, the Taï Monkey Project in Côte d'Ivoire and IVP project in South Africa, as well as Jessica Rothman, for their support and assistance. We thank the Uganda Wildlife Authority, the Uganda National Council for Science and Technology, the Ministère de la Recherche Scientifique and the Ministère de l'Agriculture et des Ressources Animales in Côte d'Ivoire and the Ezemvelo KZN Wildlife in South Africa for permission to conduct the study. We thank the editor Claudia Wascher and two anonymous reviewers for helpful comments on the manuscript.

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
