## [Reviewer comments · Royal Society Open Science]

Review History

RSOS-181275.R0 (Original submission)

Review form: Reviewer 1

Is the manuscript scientifically sound in its present form?

No

Are the interpretations and conclusions justified by the results?

No

Is the language acceptable?

Yes

Is it clear how to access all supporting data?

Yes

Do you have any ethical concerns with this paper?

No

Have you any concerns about statistical analyses in this paper?

No

Recommendation?

Major revision is needed (please make suggestions in comments)

Comments to the Author(s)

A comparative approach to vocal communication is always welcome but rarely done. Therefore, this paper looking at greeting calls in the philopatric sex is a good start. The authors study four species said to be of similar socioecology, but given that chimpanzees are considered to be a fission-fusion species and are male philopatric whereas vervets and baboons are usually classified as having stable cohesive social groups with female philopatry, can these really be considered similar in socioecology? Furthermore, the title refers to universals in greeting calls but the data do not support such a strong claim. Vervet monkey females produced no greeting calls, and low ranking animals called more than high ranking animals in the other species, but with important species differences. Target animals called more often in chimpanzees and approaching animals more in the other two species. Lower ranked animals with greater dyadic rank differences called more in chimpanzees with the opposite being true in mangabeys and little difference in baboons. This does not seem like a universal pattern.

If one were to accept that these species all have similar socioecologies, the main conclusion would appear to be "Diversity in Greeting Calls".

I can appreciate why the authors do not analyze the results for vervet monkeys since there were no greeting calls, but this species is completely missing in the Method sections describing how data were collected. No mention is made of sampling method or intensity of sampling. Does this mean that observations on vervet monkeys were casual and not systematic? Could that explain the lack of greeting calls? More needs to be presented on methods to assure a reader that strong methods were used on all species.

Why was focal animal sampling used with baboons and mangabeys but scan sampling with chimpanzees? Should not similar methods be used for better comparison?

Sex is also a confounding variable in the study. Yes, male chimpanzees are philopatric, but also have more overt aggressive interactions than female chimpanzees and females of the other species. If vervet monkeys produce greeting calls but never among females, then this also suggests a sex difference in calling in this species. The research seems incomplete without disentangling possible effects of sex from those of philopatry. The authors are likely to have gathered data on both sexes as part of their research and adding those data would make this paper stronger and provide more insights into the proximate usage and ultimate function of greeting calls.

It was also striking that greeting calls appeared so rarely. This raises questions about their function and usage that the current paper does not answer. A skeptic might question why it is of interest to pursue research on greeting calls if they are rare. Is there any evidence of what happens with a dyadic approach in the absence of greeting calls? If aggression is more likely or social interactions less likely without greeting calls, then this makes them of more interest and helps explain their role in social groups. What can the authors say about social interactions in the absence of greeting? Some further analysis of the dyadic approach data without any calling already gathered would be helpful in resolving this.

The authors criticize the approach of Laporte et al. which led to different conclusions about chimpanzees but that study focused on females so the results may not be directly comparable. The authors would have better ground for criticizing Laporte if their data on female greeting calls differed from Laporte et al.

Table 2 might be more informative if there were also summary lines excluding the silent vervet monkeys.

Figure 5 is missing axis labels.

Review form: Reviewer 2

Is the manuscript scientifically sound in its present form?

Yes

Are the interpretations and conclusions justified by the results?

No

Is the language acceptable?

Yes

Is it clear how to access all supporting data?

Yes

Do you have any ethical concerns with this paper?

No

Have you any concerns about statistical analyses in this paper?

No

Recommendation?

Accept with minor revision (please list in comments)

Comments to the Author(s)

In the paper „Universals of vocal greeting in four primate species“ the authors compare vocal greeting behaviour across four different primate species and discuss species-specific differences in the function of these calls. The paper is very well written, the introduction is clear, the method section contains all necessary information's, the results can be easily followed and I appreciate the comparative approach. In the discussion, I have problems to follow some interpretation of the results:

- 1) The authors argue that chimpanzees use the greeting call to signal submission whereas baboons and mangabeys signal friendly intent but the authors did not differentiate between the contexts of encounter or present any data on the quality of encounter. Thus, to make this statement I would expect that they use context (Agonistic versus affiliative) as an additional predictor variable in the GLMM model.
- 2) In my opinion, the result for mangabeys that low-ranking animals produce more calls than high-ranking animals (predictor = Elo-rating) contradicts the finding that high-ranking animals produce more calls to low-ranking animals (predictor = Elo-rating differences). Do you have an explanation for that? I was also wondering whether a simple classification in dominant versus subdominant encounter partner would be better than the Elo-rating score, which is in my opinion sensitive to the order of encounters. In Fig. 2 it looks that the calling probability is more affected if the Elo-rating is negative (larger decrease) than if it is positive (almost no decrease for positive differences)
- 3) I further was wondering whether the two predictor variables role and elo-differences showed an interaction. Thus, low-ranking approach more often than higher-ranking individuals?

Minor comment:

L.170: I was confused that there was no description for vervet monkeys because the information that they were excluded from the analysis was mentioned later in the manuscript.

L.176: "Calculate" is a little bit misleading, because the reader expect some formula. Maybe you

can rephrase it or add using the Elo-rating.

L.250 – 253: This was not clear. How you define interpretable terms? Is it possible to write until the main terms and significant interactions remained? When I calculate GLMMs I test the new model against the old model until there is no significant differences between both models. Did you checked whether removing terms significantly improved the model?

L.286-287: Is there a possibility to have pairwise comparisons for each species. From Fig 2 the difference for mangabeys looks not very significant to me.

L. 310: You wrote that you observed no greeting calls for vervet monkeys (L.268), but the authors already published a paper about greeting calls in this species (ref. 11). Do you have an explanation why you did not record these calls in the present study. Why did you include this species in the paper?

Fig. 1: I would add the vervets even if the mean is zero, then the reader can extract it on the first look.

Fig 2: I was wondering why there are 24 crosses for mangabeys if the authors observed only 18 individuals.

Decision letter (RSOS-181275.R0)

06-Sep-2018

Dear Dr Fedurek:

Manuscript ID RSOS-181275 entitled "Universals of vocal greeting in four primate species" which you submitted to Royal Society Open Science, has been reviewed. The comments from reviewers are included at the bottom of this letter.

In view of the criticisms of the reviewers, the manuscript has been rejected in its current form. However, a new manuscript may be submitted which takes into consideration these comments.

Please note that resubmitting your manuscript does not guarantee eventual acceptance, and that your resubmission will be subject to peer review before a decision is made.

Your resubmitted manuscript should be submitted by 06-Mar-2019. If you are unable to submit by this date please contact the Editorial Office.

Please note that Royal Society Open Science will introduce article processing charges for all new submissions received from 1 January 2018. Charges will also apply to papers transferred to Royal Society Open Science from other Royal Society Publishing journals, as well as papers submitted as part of our collaboration with the Royal Society of Chemistry (<http://rsos.royalsocietypublishing.org/chemistry>). If your manuscript is submitted and accepted for publication after 1 Jan 2018, you will be asked to pay the article processing charge, unless you request a waiver and this is approved by Royal Society Publishing. You can find out

more about the charges at <http://rsos.royalsocietypublishing.org/page/charges>. Should you have any queries, please contact openscience@royalsociety.org.

on behalf of Dr Claudia Wascher (Associate Editor) and Prof. Kevin Padian (Subject Editor)
openscience@royalsociety.org

Subject Editor Comments:

I agree that the study is potentially very interesting for us, but the reviewers and AE raise some serious concerns that may need some time to address, so a reject/resub decision will give the authors time to do that. Please take into account all comments, and thanks for your submission.

Associate Editor Comments to Author (Dr Claudia Wascher):

Associate Editor: 1

Comments to the Author:

This is an interesting paper, investigating vocal greeting behaviour in four species of primates, depending on individual features, dyadic relationships and audience effects. The authors describe species differences in calling behaviour. Calling propensity was affected by dominance rank of the focal individual and dominance relationship between initiator of an interaction and the target of an interaction. The strength of the relationship between individuals and audience composition did not affect calling behaviour. Both reviewers highlight the importance of conducting comparative approaches in vocal communication, as such the proposed contribution is an important one, however both reviewers also raise several concerns with the manuscript in its present form, which need to be addressed prior to publication. Additionally, to the reviewers' comments I would also recommend the authors to briefly summarize the direction of the main results in the abstract, rather than simply stating that they found an effect. Further, the authors selected four study species, based on the similarity in their socio-ecology as they claim, however as reviewer 1 points out, this might be an over-statement. In line with this, the authors did not record greeting vocalizations in vervet monkeys, something which according to the introduction is against the prediction, as references are provided showing that vervet monkeys do actually produce vocalisations in interactions. This begs the question why the authors could not replicate previous finding, and it is probably due to the lower frequency of encounters recorded for vervet monkey, in comparison with the frequency of encounters recorded in other species. Differences in sampling effort between species, both, in regards to frequency of interactions and also number of individuals observed are a serious concern as the study claims to compare species and this needs to be addressed prior to publication.

Reviewers' Comments to Author:

Reviewer: 1

Comments to the Author(s)

A comparative approach to vocal communication is always welcome but rarely done. Therefore, this paper looking at greeting calls in the philopatric sex is a good start. The authors study four species said to be of similar socioecology, but given that chimpanzees are considered to be a fission-fusion species and are male philopatric whereas vervets and baboons are usually classified as having stable cohesive social groups with female philopatry, can these really be

considered similar in socioecology? Furthermore, the title refers to universals in greeting calls but the data do not support such a strong claim. Vervet monkey females produced no greeting calls, and low ranking animals called more than high ranking animals in the other species, but with important species differences. Target animals called more often in chimpanzees and approaching animals more in the other two species. Lower ranked animals with greater dyadic rank differences called more in chimpanzees with the opposite being true in mangabeys and little difference in baboons. This does not seem like a universal pattern.

If one were to accept that these species all have similar socioecologies, the main conclusion would appear to be "Diversity in Greeting Calls".

I can appreciate why the authors do not analyze the results for vervet monkeys since there were no greeting calls, but this species is completely missing in the Method sections describing how data were collected. No mention is made of sampling method or intensity of sampling. Does this mean that observations on vervet monkeys were casual and not systematic? Could that explain the lack of greeting calls? More needs to be presented on methods to assure a reader that strong methods were used on all species.

Why was focal animal sampling used with baboons and mangabeys but scan sampling with chimpanzees? Should not similar methods be used for better comparison?

Sex is also a confounding variable in the study. Yes, male chimpanzees are philopatric, but also have more overt aggressive interactions than female chimpanzees and females of the other species. If vervet monkeys produce greeting calls but never among females, then this also suggests a sex difference in calling in this species. The research seems incomplete without disentangling possible effects of sex from those of philopatry. The authors are likely to have gathered data on both sexes as part of their research and adding those data would make this paper stronger and provide more insights into the proximate usage and ultimate function of greeting calls.

It was also striking that greeting calls appeared so rarely. This raises questions about their function and usage that the current paper does not answer. A skeptic might question why it is of interest to pursue research on greeting calls if they are rare. Is there any evidence of what happens with a dyadic approach in the absence of greeting calls? If aggression is more likely or social interactions less likely without greeting calls, then this makes them of more interest and helps explain their role in social groups. What can the authors say about social interactions in the absence of greeting? Some further analysis of the dyadic approach data without any calling already gathered would be helpful in resolving this.

The authors criticize the approach of Laporte et al. which led to different conclusions about chimpanzees but that study focused on females so the results may not be directly comparable. The authors would have better ground for criticizing Laporte if their data on female greeting calls differed from Laporte et al.

Table 2 might be more informative if there were also summary lines excluding the silent vervet monkeys.

Figure 5 is missing axis labels.

Reviewer: 2

Comments to the Author(s)

In the paper „Universals of vocal greeting in four primate species“ the authors compare vocal greeting behaviour across four different primate species and discuss species-specific differences in the function of these calls. The paper is very well written, the introduction is clear, the method section contains all necessary information's, the results can be easily followed and I appreciate the comparative approach. In the discussion, I have problems to follow some interpretation of the results:

1) The authors argue that chimpanzees use the greeting call to signal submission whereas baboons and mangabeys signal friendly intent but the authors did not differentiate between the

contexts of encounter or present any data on the quality of encounter. Thus, to make this statement I would expect that they use context (Agonistic versus affiliative) as an additional predictor variable in the GLMM model.

2) In my opinion, the result for mangabeys that low-ranking animals produce more calls than high-ranking animals (predictor = Elo-rating) contradicts the finding that high-ranking animals produce more calls to low-ranking animals (predictor = Elo-rating differences). Do you have an explanation for that? I was also wondering whether a simple classification in dominant versus subdominant encounter partner would be better than the Elo-rating score, which is in my opinion sensitive to the order of encounters. In Fig. 2 it looks that the calling probability is more affected if the Elo-rating is negative (larger decrease) than if it is positive (almost no decrease for positive differences)

3) I further was wondering whether the two predictor variables role and elo-differences showed an interaction. Thus, low-ranking approach more often than higher-ranking individuals?

Minor comment:

L.170: I was confused that there was no description for vervet monkeys because the information that they were excluded from the analysis was mentioned later in the manuscript.

L.176: "Calculate" is a little bit misleading, because the reader expect some formula. Maybe you can rephrase it or add using the Elo-rating.

L.250 - 253: This was not clear. How you define interpretable terms? Is it possible to write until the main terms and significant interactions remained? When I calculate GLMMs I test the new model against the old model until there is no significant differences between both models. Did you checked whether removing terms significantly improved the model?

L.286-287: Is there a possibility to have pairwise comparisons for each species. From Fig 2 the difference for mangabeys looks not very significant to me.

L. 310: You wrote that you observed no greeting calls for vervet monkeys (L.268), but the authors already published a paper about greeting calls in this species (ref. 11). Do you have an explanation why you did not record these calls in the present study. Why did you include this species in the paper?

Fig. 1: I would add the vervets even if the mean is zero, then the reader can extract it on the first look.

Fig 2: I was wondering why there are 24 crosses for mangabeys if the authors observed only 18 individuals.

Author's Response to Decision Letter for (RSOS-181275.R0)

See Appendix A.

RSOS-182181.R0

Review form: Reviewer 1

Is the manuscript scientifically sound in its present form?

Yes

Are the interpretations and conclusions justified by the results?

Yes

Is the language acceptable?

Yes

Is it clear how to access all supporting data?

Yes

Do you have any ethical concerns with this paper?

No

Have you any concerns about statistical analyses in this paper?

I do not feel qualified to assess the statistics

Recommendation?

Accept with minor revision (please list in comments)

Comments to the Author(s)

The original premise of this paper was that among species with similar socioecology, greeting calls among members of the philopatric sex would be similar across species. The original reviews questioned whether the species were in fact similar in social ecology and noted that there were many differences among the four species studied. The authors have addressed these concerns quite well in their revision and now emphasize that their work focuses mainly on proximate mechanisms rather than on functional aspects. All of this is quite good and the manuscript now more accurately reflects what was done and what results were obtained. However, the points that would have made this an exciting paper suitable for a Royal Society publication are no longer present. The current manuscript would make a fine contribution to a high quality specialized journal on primate behavior and ecology, but I'm not sure it will have the broad appeal that is typically looked for by the Royal Society, although I gather that is not relevant to Open Science. I'll leave it to the action editor to make that call.

Minor Points:

- l. 80 use "whereas" or "although" rather than "while"
- l. 120, "whereas" not "while"
- l. 226, all animals or just males, since males are the focus.
- l. 247 all animals or all females?
- l. 325 define LRT on first mention
- l. 364 "whereas" not "while"
- l. 497 after arguing that the paper is not interested in function, function is now introduced.
- l. 523 can you give some examples of which types of species would be most fruitful for future research and why?

References: Should not Latin binomials be in italics?

Review form: Reviewer 3**Is the manuscript scientifically sound in its present form?**

Yes

Are the interpretations and conclusions justified by the results?

Yes

Is the language acceptable?

Yes

Is it clear how to access all supporting data?

No

Do you have any ethical concerns with this paper?

Yes

Have you any concerns about statistical analyses in this paper?

No

Recommendation?

Accept with minor revision (please list in comments)

Comments to the Author(s)

Authors addressed almost all requests of former referees. Regarding the revision, there were some minors, which I will outline below.

1. Title. Authors modified the title and the introduction in the revision to fulfill the requirements of the former referees. According to my view, the modified title implies that authors will address proximate mechanisms of vocal production, e.g. neural, hormonal, genetic, developmental networks or modifications of the peripheral vocal tract affecting vocal production. However, this is not what authors have studied in the MS. They studied the occurrence of vocalizations, given during approach behavior in dyads of the philopatric sex in a social group of four species of nonhuman primates and consider thereby the rank of the focal animal, dominance relationships and social roles within dyads, and audience effects, see also line 81 ff. It is on the Editor to decide whether the modified title reflects what authors mean.

2. "Greeting" calls are functionally defined calls. In this MS, authors term vocalizations, given during approach behavior of a focal animal "Greeting" calls. The function of these calls, however, is not yet known for the respective species and according to authors will not be addressed in this MS. Such a situation makes a comparison across species tricky since you may compare then apples with oranges.

3. Abstract. Line 36: dominance rank. Do you mean dominance or rank order?

4. In the Abstract, authors use the term "dyadic approach". Please address in the methods the link between dyadic approach and an "encounter".

5. Introduction. Line 74. Please embed Scheumann et al. 2017 to refer to current knowledge on how primate "greeting" calls can be defined and studied (Scheumann, M.; Linn, S.; Zimmermann, E.: Vocal greeting during mother-infant reunions in a nocturnal primate, the gray mouse lemur (*Microcebus murinus*). In: Scientific Reports 7 (2017) 10321)

6. Line 113: Please give reference for the divergence in the acoustic structure of "greeting" calls between the four studied species.

7. Methods. 157 ff and Discussion. As one of the former referees mentioned, not the same methods were applied for all species. This is still the case in the revised MS. Please outline in the discussion how this situation may have shaped your findings. Further, one of the referees mentioned that authors did not present any data on the context of an encounter or the quality of an encounter. This issue remains in the revised MS and thus should be addressed in the discussion.

8. Line 258. Please outline the concept of Elo-rating here so that the general audience will be well informed upon the procedure you used here for comparative purposes. To refer to a reference is not sufficient to understand the content of your paper.

9. Abstract/Results: I still share the opinion of referee 2 that the following findings for mangabeys are difficult to understand: "low ranking animals were more likely to call than high-ranking ones" and "calling being negatively associated with rank distance" (see e.g. Abstract). Please, address this issue in the discussion to make your findings better understandable for the general audience.

10. Please check Figure 1, 2 and 5. In my version, I cannot see the description on the y-axis.
11. Please address whether all the primary data supporting the findings of the paper are publicly available.
12. Please give research permission no. in the paragraph Ethics.

Decision letter (RSOS-182181.R0)

07-Feb-2019

Dear Dr Fedurek,

The Subject Editor assigned to your paper ("Patterns of vocal greeting production in four primate species") has now received comments from reviewers. We would like you to revise your paper in accordance with the referee and Associate Editor suggestions which can be found below (not including confidential reports to the Editor). Please note this decision does not guarantee eventual acceptance.

Please submit a copy of your revised paper before 02-Mar-2019. Please note that the revision deadline will expire at 00.00am on this date. If we do not hear from you within this time then it will be assumed that the paper has been withdrawn. In exceptional circumstances, extensions may be possible if agreed with the Editorial Office in advance. We do not allow multiple rounds of revision so we urge you to make every effort to fully address all of the comments at this stage. If deemed necessary by the Editors, your manuscript will be sent back to one or more of the original reviewers for assessment. If the original reviewers are not available we may invite new reviewers.

When submitting your revised manuscript, you must respond to the comments made by the referees and upload a file "Response to Referees" in "Section 6 - File Upload". Please use this to document how you have responded to each of the comments, and the adjustments you have made. In order to expedite the processing of the revised manuscript, please be as specific as possible in your response.

- Ethics statement

- Data accessibility

It is a condition of publication that all supporting data are made available either as supplementary information or preferably in a suitable permanent repository. The data accessibility section should state where the article's supporting data can be accessed. This section

should also include details, where possible of where to access other relevant research materials such as statistical tools, protocols, software etc can be accessed. If the data has been deposited in an external repository this section should list the database, accession number and link to the DOI for all data from the article that has been made publicly available. Data sets that have been deposited in an external repository and have a DOI should also be appropriately cited in the manuscript and included in the reference list.

If you wish to submit your supporting data or code to Dryad (<http://datadryad.org/>), or modify your current submission to dryad, please use the following link:
<http://datadryad.org/submit?journalID=RSOS&manu=RSOS-182181>

- **Competing interests**

- **Authors' contributions**

- **Acknowledgements**

- **Funding statement**

on behalf of Dr Claudia Wascher (Associate Editor) and Professor Kevin Padian (Subject Editor)
openscience@royalsociety.org

Associate Editor Comments to Author (Dr Claudia Wascher):

Thank you very much for submitting a revised version of your manuscript. I have now received reports from both original reviewers and both acknowledge that their previous comments have been addressed. However, they raise concerns that the focus has significantly shifted from the first version of the manuscript. I do not necessarily believe that this made the manuscript unsuitable for Royal Society Open Science. The journal aims at publishing manuscripts that are considered scientifically sound and sufficiently advance scientific knowledge. The judgement as to importance and significance is left to the individual reader. The reviewers have confirmed that your contribution is scientifically sound and I do believe that your manuscript has the potential to advance scientific knowledge, as the focus on proximate mechanisms of behaviour is often underrepresented. In this respect I believe your manuscript could make an important contribution. I am afraid that at the present stage, the focus of the manuscript is not entirely clear. I do not recognise what the advantage of a comparative approach to study mechanisms of vocal greetings, in comparison to study these aspects in a single species? Did you have specific predictions that, for some characteristics in their ecology and/or social system, the species' would respond similar or different? If you do not have clear predictions with regards to the species compared, it might be worth to shift the focus away from it even more. In your introduction you presently state that studying mechanisms of behaviour is important and comparing species is important, but you do not explain to the reader why this is important and what we can learn from it, e.g. why is it important to understand how relationships between individuals affect vocal behaviour? You very specifically describe the details about what you studied, but you do not explain why this is important, e.g. why the different levels of analysis matter. You nicely describe what individual species do with regards to vocal behaviour, but you do not explain what we learn from this about the importance of the behaviour. Parts of this is already present in the discussion but it can certainly be highlighted more in the introduction.

I hope you find the detailed comments made by the reviewers and my overall comments useful when preparing a revision of your manuscript.

Reviewer comments to Author:

Reviewer: 1

Comments to the Author(s)

The original premise of this paper was that among species with similar socioecology, greeting calls among members of the philopatric sex would be similar across species. The original reviews questioned whether the species were in fact similar in social ecology and noted that there were many differences among the four species studied. The authors have addressed these concerns quite well in their revision and now emphasize that their work focuses mainly on proximate mechanisms rather than on functional aspects. All of this is quite good and the manuscript now more accurately reflects what was done and what results were obtained. However, the points that would have made this an exciting paper suitable for a Royal Society publication are no longer present. The current manuscript would make a fine contribution to a high quality specialized journal on primate behavior and ecology, but I'm not sure it will have the broad appeal that is typically looked for by the Royal Society, although I gather that is not relevant to Open Science. I'll leave it to the action editor to make that call.

Minor Points:

- l. 80 use "whereas" or "although" rather than "while"
- l. 120, "whereas" not "while"
- l. 226, all animals or just males, since males are the focus.

1. 247 all animals or all females?
 1. 325 define LRT on first mention
 1. 364 “whereas” not “while”
 1. 497 after arguing that the paper is not interested in function, function is now introduced.
 1. 523 can you give some examples of which types of species would be most fruitful for future research and why?
- References: Should not Latin binomials be in italics?

Reviewer: 3

Comments to the Author(s)

Authors addressed almost all requests of former referees. Regarding the revision, there were some minors, which I will outline below.

1. Title. Authors modified the title and the introduction in the revision to fulfill the requirements of the former referees. According to my view, the modified title implies that authors will address proximate mechanisms of vocal production, e.g. neural, hormonal, genetic, developmental networks or modifications of the peripheral vocal tract affecting vocal production. However, this is not what authors have studied in the MS. They studied the occurrence of vocalizations, given during approach behavior in dyads of the philopatric sex in a social group of four species of nonhuman primates and consider thereby the rank of the focal animal, dominance relationships and social roles within dyads, and audience effects, see also line 81 ff. It is on the Editor to decide whether the modified title reflects what authors mean.

2. “Greeting” calls are functionally defined calls. In this MS, authors term vocalizations, given during approach behavior of a focal animal “Greeting” calls. The function of these calls, however, is not yet known for the respective species and according to authors will not be addressed in this MS. Such a situation makes a comparison across species tricky since you may compare then apples with oranges.

3. Abstract. Line 36: dominance rank. Do you mean dominance or rank order?

4. In the Abstract, authors use the term “dyadic approach”. Please address in the methods the link between dyadic approach and an “encounter”.

5. Introduction. Line 74. Please embed Scheumann et al. 2017 to refer to current knowledge on how primate “greeting” calls can be defined and studied (Scheumann, M.; Linn, S.; Zimmermann, E.: Vocal greeting during mother-infant reunions in a nocturnal primate, the gray mouse lemur (*Microcebus murinus*). In: *Scientific Reports* 7 (2017) 10321)

6. Line 113: Please give reference for the divergence in the acoustic structure of “greeting” calls between the four studied species.

7. Methods. 157 ff and Discussion. As one of the former referees mentioned, not the same methods were applied for all species. This is still the case in the revised MS. Please outline in the discussion how this situation may have shaped your findings. Further, one of the referees mentioned that authors did not present any data on the context of an encounter or the quality of an encounter. This issue remains in the revised MS and thus should be addressed in the discussion.

8. Line 258. Please outline the concept of Elo-rating here so that the general audience will be well informed upon the procedure you used here for comparative purposes. To refer to a reference is not sufficient to understand the content of your paper.

9. Abstract/Results: I still share the opinion of referee 2 that the following findings for mangabeys are difficult to understand: “low ranking animals were more likely to call than high-ranking ones” and “calling being negatively associated with rank distance ” (see e.g. Abstract). Please, address this issue in the discussion to make your findings better understandable for the general audience.

10. Please check Figure 1, 2 and 5. In my version, I cannot see the description on the y-axis.

11. Please address whether all the primary data supporting the findings of the paper are publicly available.
12. Please give research permission no. in the paragraph Ethics.

Author's Response to Decision Letter for (RSOS-182181.R0)

See Appendix B.

Decision letter (RSOS-182181.R1)

21-Mar-2019

Dear Dr Fedurek,

I am pleased to inform you that your manuscript entitled "Behavioural patterns of vocal greeting production in four primate species" is now accepted for publication in Royal Society Open Science.

on behalf of Dr Claudia Wascher (Associate Editor) and Professor Kevin Padian (Subject Editor)
openscience@royalsociety.org

Appendix A

We thank the Editor and the Reviewers for their helpful comments. We have addressed, or responded to, all the comments and suggestions. As a result, we believe that the manuscript has improved considerably. We provide detailed responses to particular comments below (in red).

Associate Editor Comments to Author

This is an interesting paper, investigating vocal greeting behaviour in four species of primates, depending on individual features, dyadic relationships and audience effects. The authors describe species differences in calling behaviour. Calling propensity was affected by dominance rank of the focal individual and dominance relationship between initiator of an interaction and the target of an interaction. The strength of the relationship between individuals and audience composition did not affect calling behaviour. Both reviewers highlight the importance of conducting comparative approaches in vocal communication, as such the proposed contribution is an important one, however both reviewers also raise several concerns with the manuscript in its present form, which need to be addressed prior to publication.

We thank the editor for these encouraging and thoughtful remarks. We provide point-by-point responses below.

Additionally, to the reviewers' comments I would also recommend the authors to briefly summarize the direction of the main results in the abstract, rather than simply stating that they found an effect.

We have now incorporated a brief summary of the results in the Abstract (lines 41-44).

Further, the authors selected four study species, based on the similarity in their socio-ecology as they claim, however as reviewer 1 points out, this might be an over-statement.

This is a good point. We have removed this part.

In line with this, the authors did not record greeting vocalizations in vervet monkeys, something which according to the introduction is against the prediction, as references are provided showing that vervet monkeys do

actually produce vocalisations in interactions. This begs the question why the authors could not replicate previous finding, and it is probably due to the lower frequency of encounters recorded for vervet monkey, in comparison with the frequency of encounters recorded in other species.

The study by Mercier et al (2017) is the only one that studied systematically greeting calls in vervet monkeys. That study looked at both female-female and female-male interactions, whereas our study focuses only on female-female interactions (we only looked at the philopatric sex for reasons explained below). During encounters females produce grunts exclusively towards males as shown in Mercier et al 2017, while females do not produce calls towards other females (lines 335-337). We have now made the differences between the two studies more explicit in the text (lines 335-337). As explained below, we provide data only on one sex (the philopatric sex) for each species because not for all of our study species we have data from both sexes. The advantage of having data from the philopatric sex (as opposed to non-philopatric) is that members of this sex can form long-lasting social bonds, which is not always the case with the non-philopatric sex (mainly due to frequent migrations of members of this sex, for example in vervets).

Differences in sampling effort between species, both, in regards to frequency of interactions and also number of individuals observed are a serious concern as the study claims to compare species and this needs to be addressed prior to publication.

We appreciate this comment and we agree that compared to the other three species, we have a smaller number of observations from vervets. While we agree that ideally the vervet numbers were comparable with those from the other three species, we believe that these numbers are sufficient to show that vervets typically do not call when approaching another female (i.e. we observed no calls during 61 encounters). Therefore, a higher number of data points from this species would likely not change our conclusions.

Reviewers' Comments to Author:

Reviewer: 1

Comments to the Author(s)

A comparative approach to vocal communication is always welcome but rarely done. Therefore, this paper looking at greeting calls in the philopatric sex is a good start. The authors study four species said to be of similar

socioecology, but given that chimpanzees are considered to be a fission-fusion species and are male philopatric whereas vervets and baboons are usually classified as having stable cohesive social groups with female philopatry, can these really be considered similar in socioecology?

We agree that we overgeneralized our statement and we have reworded this part accordingly.

Furthermore, the title refers to universals in greeting calls but the data do not support such a strong claim. Vervet monkey females produced no greeting calls, and low ranking animals called more than high ranking animals in the other species, but with important species differences. Target animals called more often in chimpanzees and approaching animals more in the other two species. Lower ranked animals with greater dyadic rank differences called more in chimpanzees with the opposite being true in mangabeys and little difference in baboons. This does not seem like a universal pattern. If one were to accept that these species all have similar socioecologies, the main conclusion would appear to be "Diversity in Greeting Calls".

We agree and have reworded the title to avoid the problem highlighted by the reviewer: "Patterns of vocal greeting production in four primate species". We have also removed from the text the part relating to socioecology similarities.

I can appreciate why the authors do not analyze the results for vervet monkeys since there were no greeting calls, but this species is completely missing in the Method sections describing how data were collected. No mention is made of sampling method or intensity of sampling. Does this mean that observations on vervet monkeys were casual and not systematic? Could that explain the lack of greeting calls? More needs to be presented on methods to assure a reader that strong methods were used on all species.

This is a good point and we thank the reviewer for pointing out this omission. We now provide the information on how data on vervets were collected in the Methods (lines 209-216).

Why was focal animal sampling used with baboons and mangabeys but scan sampling with chimpanzees? Should not similar methods be used for better comparison?

The core data that were compared across the four species, the encounter data, were recorded using the same sampling method (focal animals sampling). Only the methods for establishing friendships differed. For chimpanzees, for example, we used scan sampling based on proximity scores, which as an established method for this species and field site. While the data collection regime differed, the higher-level concepts we assessed can reasonably be assumed to be measuring the same aspects (e.g. rank difference, bond strength).

Sex is also a confounding variable in the study. Yes, male chimpanzees are philopatric, but also have more overt aggressive interactions than female chimpanzees and females of the other species. If vervet monkeys produce greeting calls but never among females, then this also suggests a sex difference in calling in this species. The research seems incomplete without disentangling possible effects of sex from those of philopatry. The authors are likely to have gathered data on both sexes as part of their research and adding those data would make this paper stronger and provide more insights into the proximate usage and ultimate function of greeting calls.

We very much appreciate this comment. However, we did not systematically collect focal animal data on both sexes in all four species. Hence, unfortunately, we are unable to comply with the reviewer's request. Instead, we present data on interactions between members of the philopatric sex for which we have adequate amount of data for each species.

The advantage of looking at the philopatric sex, as opposed to the non-philopatric sex, is that dominance and affiliative relationships within this sex are clearly established and easily measurable mainly due to infrequent migrations of individuals within this sex category.

Regarding the ultimate functions of calls, although we agree with the Reviewer that this is an interesting research avenue, we would like to point out that this was not the aim of this study. We have also reworded parts in the Introduction (lines 130-132) and the Discussion (lines 400-403, 435-437) to make it clear that exploring ultimate functions of greeting calls was not our aim. We have also changed the title of the paper, which now also suggests that we are only concerned with factors mediating vocal greeting production rather than with the function of calling.

It was also striking that greeting calls appeared so rarely. This raises questions about their function and usage that the current paper does not

answer. A skeptic might question why it is of interest to pursue research on greeting calls if they are rare. Is there any evidence of what happens with a dyadic approach in the absence of greeting calls? If aggression is more likely or social interactions less likely without greeting calls, then this makes them of more interest and helps explain their role in social groups. What can the authors say about social interactions in the absence of greeting? Some further analysis of the dyadic approach data without any calling already gathered would be helpful in resolving this.

This is a good point. One reason for the apparent rarity of greeting calls is that we considered a fairly wide range of scenarios where a greeting **could have happened**. That is, we collected data on approaches. Most of these approaches evidently did not lead to any greeting and interaction and hence the “rate” appears to be low. But we believe that this is the correct approach to investigate the factors that drive whether or not a greeting actually occurs conditional on the occurrence of an approach.

Furthermore, we are sympathetic to the question about the function of greeting calls, i.e. what consequences do calls have, if they occur, on any actual subsequent social interaction. But this, as mentioned above, goes much beyond what this study intended, i.e. looking into the mechanisms of vocal greeting production. We believe that our aim is now reflected better in the new title of the paper: “Patterns of vocal greeting production in four primate species”.

The authors criticize the approach of Laporte et al. which led to different conclusions about chimpanzees but that study focused on females so the results may not be directly comparable. The authors would have better ground for criticizing Laporte if their data on female greeting calls differed from Laporte et al.

We agree with this statement and mention this possibility in the Discussion (lines 418-420). However, our criticism of Laporte’s study is actually more general than that: this study makes claims about audience effects, but ignores some evidently simpler factors that might account for call production, i.e. dyadic features and features of the caller. As such, it is less relevant that this study was done on females. We have amended this section so that it better illustrates the more general point we want to make here (lines 412-418).

Table 2 might be more informative if there were also summary lines excluding the silent vervet monkeys.

Although we appreciate this comment, we believe that providing two sets of summaries (with and without vervets) could be confusing and not consistent with the way we present data from the four species elsewhere (e.g. Fig 1). But we can change this if the Reviewer or Editor insists.

Figure 5 is missing axis labels.

Thank you for spotting this. We now attach the correct figure.

Reviewer: 2

Comments to the Author(s)

In the paper „Universals of vocal greeting in four primate species“ the authors compare vocal greeting behaviour across four different primate species and discuss species-specific differences in the function of these calls. The paper is very well written, the introduction is clear, the method section contains all necessary information's, the results can be easily followed and I appreciate the comparative approach.

We thank the reviewer for these positive comments.

In the discussion, I have problems to follow some interpretation of the results:

- 1) The authors argue that chimpanzees use the greeting call to signal submission whereas baboons and mangabeys signal friendly intent but the authors did not differentiate between the contexts of encounter or present any data on the quality of encounter. Thus, to make this statement I would expect that they use context (Agonistic versus affiliative) as an additional predictor variable in the GLMM model.

We would like to note that investigating the function of greeting calls (e.g. affiliative versus submissive) was not the aim of this paper since we only looked at the mechanism mediating the production of these calls. With regard to the suggestion to include in our models “context” as another variable, this would be tricky since interactions (whether aggressive or affiliative) follow rather than precede greetings. We thus believe that including such pre-defined contexts would not be feasible for our analyses and would invalidate the temporal sequence of events (i.e. context manifests only after the approach). In addition, it would be difficult to assign a context to those encounters that are not followed by an interaction. And again,

incorporating the context of calls would be more relevant if we dealt with their ultimate functions of calls, which however is not the case.

Nonetheless, we agree with the Reviewer that the way we refer in the text to the function of greeting calls might be confusing. We have now reworded these parts of the Introduction and the Discussion so as to avoid this confusion. We now also state in the Introduction (lines 130-132) and the Discussion (lines 400-403, 435-437) that the purpose of this study was not exploring the function of greeting calls. For the same reason we have also amended the title of the paper.

2) In my opinion, the result for mangabeys that low-ranking animals produce more calls than high-ranking animals (predictor = Elo-rating) contradicts the finding that high-ranking animals produce more calls to low-ranking animals (predictor = Elo-rating differences). Do you have an explanation for that? I was also wondering whether a simple classification in dominant versus subdominant encounter partner would be better than the Elo-rating score, which is in my opinion sensitive to the order of encounters. In Fig. 2 it looks that the calling probability is more affected if the Elo-rating is negative (larger decrease) than if it is positive (almost no decrease for positive differences).

We think the first part of this comment is a misunderstanding. Elo-rating difference quantifies magnitude and sign of a rating difference and as such is a dyadic feature. For example, an individual with a low rating (and hence a relatively high probability to call) can still, in addition to this, increase its calling propensity even more if the approached individual is even lower rated. As such, there is no contradiction here.

We also refrained from using binary status because Elo-rating provides much more detailed information and often provides a better approximation of competitive abilities (e.g. Neumann et al 2018, Anim Behav). Of course the reviewer is correct in that Elo-rating is dependent on the temporal sequence of interactions, but this is a design feature of the method and not a drawback. Using an ordinal ranking method instead would come with its own set of problems and we chose Elo-rating because it assumes neither transitivity nor linearity of the dominance network, but instead provides a cardinal measure of competitive ability that usually correlates closely with “classic” dominance ranks (Neumann et al 2011, 2018).

3) I further was wondering whether the two predictor variables role and elo-differences showed an interaction.

There was no significant interaction between these variables. This interaction was considered in the full model and removed because it was not significant (table 2). We added a new supplemental figure (figure S1) that shows graphically the results of the full model. The interaction the reviewer is interested in is displayed in the second row of this figure.

Thus, low-ranking approach more often than higher-ranking individuals?

This is an interesting question but this was not the aim of this study: we did not explore variables that explain who is more likely to approach who, but who is more likely to call when approaching, or being approach, by another individual. Out of curiosity, we fitted an initial model to address this question. The results of this model suggest that only for baboons low-rated individuals are more likely to be approacher compared to high-rated individuals. For chimpanzees and mangabeys, the opposite seems to be the case (see attached figure). However, we would like to note here that in addition to being not directly relevant for our study, this is a very superficial analysis and almost certainly not the ideal way of addressing the question with regard to who **approaches** more often. This is because this analysis is conditional on an approach having occurred in the first place and hence it is uninformative about rates (i.e. approaches per time).

Minor comment:

L.170: I was confused that there was no description for vervet monkeys because the information that they were excluded from the analysis was mentioned later in the manuscript.

We thank the Reviewer you for spotting this omission. Reviewer 1 also alerted us about this. We now include the information on how data on vervets were collected (lines 209-216).

L.176: “Calculate” is a little bit misleading, because the reader expect some formula. Maybe you can rephrase it or add using the Elo-rating.

We have rephrased this into “establishes”. We don’t provide formulas because these appear in the cited references in the section that actually states that we used Elo-rating for assigning dominance scores.

L.250 – 253: This was not clear. How you define interpretable terms? Is it possible to write until the main terms and significant interactions remained?

When I calculate GLMMs I test the new model against the old model until there is no significant differences between both models. Did you checked whether removing terms significantly improved the model?

We are not sure we understand this comment correctly. We defined interpretable terms (in a model) as “a model with interpretable terms, i.e., with significant interaction terms and/or main effects (either significant or non-significant)” (l. 251f in old version). The crucial aspect for us in this approach is that this model simplification is conditional on the comparison between full (i.e. initial) and null model being significant. In this sense, we compared models by removing **interaction terms**. To make this clearer, we added the phrase “using LRTs [likelihood ratio tests]” when describing the simplification procedure and also added some more references here. We also wish to point out that the procedure is included in the supplementary R script and the interested reader can reproduce our steps using the information presented there. We also included an explicit reference to our code in this section.

L.286-287: Is there a possibility to have pairwise comparisons for each species. From Fig 2 the difference for mangabeys looks not very significant to me.

We apologize for having submitted a wrong figure 2. The correct version is in the current submission. In response to the reviewer’s comment, we agree that the mangabey data appear not to be in line with the general trend in this figure. There are two points related to this. First, symbols in the figure represent raw data and as such do not necessarily represent the general pattern once all other variables are controlled for. Second, what the Reviewer seems to suggest (different slopes for the different species) is akin to testing the interaction between species and Elo-rating. We have done this during our model simplification procedure and this interaction term was not significant and hence removed. The new supplementary figure (figure S1) illustrated this (in the top row, all slopes are negative for all species and for both targets and approachers). It therefore seems prudent to conclude that based on our data the negative effect of Elo-rating on calling probability is a simple main effect.

L. 310: You wrote that you observed no greeting calls for vervet monkeys (L.268), but the authors already published a paper about greeting calls in this species (ref. 11). Do you have an explanation why you did not record these calls in the present study. Why did you include this species in the paper?

Mercier et al (2017) study looked at both female-female and female-male interactions while our study focuses only on female-female interactions (we only looked at the philopatric sex for reasons explained above). While during encounters females produce grunts predominantly to males as shown in Mercier et al 2017, females do not produce calls towards other females (lines 335-337) (see table A5 in Mercier et al 2017). We agree this might be confusing so we have now made the differences between the two studies more explicit in the text (lines 335-337).

Fig. 1: I would add the vervets even if the mean is zero, then the reader can extract it on the first look.

We agree. We have amended this figure accordingly.

Fig 2: I was wondering why there are 24 crosses for mangabeys if the authors observed only 18 individuals.

Thank you for alerting us about this mistake. We now provide the correct figure.

References:

Mercier, S., Neumann, C., van de Waal, E., Chollet, E., Meric de Bellefon, J., & Zuberbühler, K. (2017). Vervet monkeys greet adult males during high-risk situations. *Animal Behaviour*, *132*, 229–245.

<https://doi.org/10.1016/j.anbehav.2017.07.021>

Neumann, C., Duboscq, J., Dubuc, C., Ginting, A., Irwan, A. M., Agil, M., ... Engelhardt, A. (2011). Assessing dominance hierarchies: validation and advantages of progressive evaluation with Elo-rating. *Animal Behaviour*, *82*, 911–921. <https://doi.org/10.1016/j.anbehav.2011.07.016>

Neumann, C., McDonald, D. B., & Shizuka, D. (2018). Dominance ranks, dominance ratings and linear hierarchies: a critique. *Animal Behaviour*, *144*, e1–e16. <https://doi.org/10.1016/j.anbehav.2018.07.012>

Appendix B

We thank the Editor and the Reviewers for their useful comments. We believe we have addressed all the issues raised resulting in a considerably improved manuscript. Please find our responses below particular comments.

Associate Editor Comments to Author (Dr Claudia Wascher):
Thank you very much for submitting a revised version of your manuscript. I have now received reports from both original reviewers and both acknowledge that their previous comments have been addressed. However, they raise concerns that the focus has significantly shifted from the first version of the manuscript. I do not necessarily believe that this made the manuscript unsuitable for Royal Society Open Science. The journal aims at publishing manuscripts that are considered scientifically sound and sufficiently advance scientific knowledge. The judgement as to importance and significance is left to the individual reader. The reviewers have confirmed that your contribution is scientifically sound and I do believe that your manuscript has the potential to advance scientific knowledge, as the focus on proximate mechanisms of behaviour is often underrepresented. In this respect I believe your manuscript could make an important contribution.

Thank you.

I am afraid that at the present stage, the focus of the manuscript is not entirely clear. I do not recognise what the advantage of a comparative approach to study mechanisms of vocal greetings, in comparison to study these aspects in a single species? Did you have specific predictions that, for some characteristics in their ecology and/or social system, the species' would respond similar or different? If you do not have clear predictions with regards to the species compared, it might be worth to shift the focus away from it even more.

We appreciate the Editor's criticism and suggestion. We have elaborated on the rationale behind our comparative approach (L 87-102). As we outline in the Introduction, our working hypothesis is that it appears plausible for some greeting mechanisms to be shared across species while others differ

between species (L 77-87). This leads to the paper being by and large explorative rather than driven by specific hypotheses, not the least because we lack very much any prior data for at least two of the four species (in fact, also our knowledge on olive baboons is largely based on findings of a closely related species, which we now make more explicit). Furthermore, we do not know of any study that incorporated all the variables that we included simultaneously. In addition to this, we find it challenging to add predictions at this stage of the study because we now know the results of our analyses, and this poses an ethical conundrum to us. Hence, it would be fairly easy to just fall prey to confirmation bias such that we come up with predictions that are ultimately met.

In your introduction you presently state that studying mechanisms of behaviour is important and comparing species is important, but you do not explain to the reader why this is important and what we can learn from it, e.g. why is it important to understand how relationships between individuals affect vocal behaviour? You very specifically describe the details about what you studied, but you do not explain why this is important, e.g. why the different levels of analysis matter.

We agree and have added parts in the introduction to highlight advantages of the comparative approach that we adopt (L 92-102).

You nicely describe what individual species do with regards to vocal behaviour, but you do not explain what we learn from this about the importance of the behaviour. Parts of this is already present in the discussion but it can certainly be highlighted more in the introduction.

Again, we believe that making the Introduction more hypothesis-driven is not appropriate for the reasons outlined above. Instead, we relate our findings to what we know about the greeting behaviour in the four species in the Discussion. However, as mentioned above, we have followed the Editor's suggestion and elaborated in the Introduction the rationale behind our comparative approach (L 92-102).

I hope you find the detailed comments made by the reviewers and my overall comments useful when preparing a revision of your manuscript.

Reviewer comments to Author:

Reviewer: 1

Comments to the Author(s)

The original premise of this paper was that among species with similar socioecology, greeting calls among members of the philopatric sex would be similar across species. The original reviews questioned whether the species were in fact similar in social ecology and noted that there were many differences among the four species studied. The authors have addressed these concerns quite well in their revision and now emphasize that their work focuses mainly on proximate mechanisms rather than on functional aspects. All of this is quite good and the manuscript now more accurately reflects what was done and what results were obtained. However, the points that would have made this an exciting paper suitable for a Royal Society publication are no longer present. The current manuscript would make a fine contribution to a high quality specialized journal on primate behavior and ecology, but I'm not sure it will have the broad appeal that is typically looked for by the Royal Society, although I gather that is not relevant to Open Science. I'll leave it to the action editor to make that call.

We thank the Reviewer for its positive comments on the manuscript. Following the Reviewer's, but also the Editor's comments, we have reconstructed the Introduction a bit to justify our comparative approach (L 92-102).

Minor Points:

l. 80 use "whereas" or "although" rather than "while"

We have done this (L 75).

l. 120, "whereas" not "while"

We have corrected this (L 129).

l. 226, all animals or just males, since males are the focus.

Thank you for spotting this. We have corrected this (L 234).

l. 247 all animals or all females?

We have corrected this (L 252).

l. 325 define LRT on first mention

The reviewer probably overlooked it, but LRT is defined just prior to the first use of the abbreviation as likelihood ratio test (L 335).

l. 364 “whereas” not “while”

We have corrected this (L 370).

l. 497 after arguing that the paper is not interested in function, function is now introduced.

Although the aim of this study is not to investigate the function of these calls, it is sometimes challenging to present these results without relating to the function attributed to these calls because that is what most published findings are concerned with. But we have made it clear in the Introduction that while we refer to the function of call, investigating the function of these calls is not the aim of this paper (L 161-165).

l. 523 can you give some examples of which types of species would be most fruitful for future research and why?

We agree with the Reviewer and have elaborated on this (L 538-540).

References: Should not Latin binomials be in italics?

We have amended this.

Reviewer: 3

Comments to the Author(s)

Authors addressed almost all requests of former referees.

Regarding the revision, there were some minors, which I will

outline below.

1. Title. Authors modified the title and the introduction in the revision to fulfill the requirements of the former referees. According to my view, the modified title implies that authors will address proximate mechanisms of vocal production, e.g. neural, hormonal, genetic, developmental networks or modifications of the peripheral vocal tract affecting vocal production. However, this is not what authors have studied in the MS. They studied the occurrence of vocalizations, given during approach behavior in dyads of the philopatric sex in a social group of four species of nonhuman primates and consider thereby the rank of the focal animal, dominance relationships and social roles within dyads, and audience effects, see also line 81 ff. It is on the Editor to decide whether the modified title reflects what authors mean.

We have modified the title of the paper to address the Reviewer's concerns and to make it clear that we explore only behavioural patterns of vocal greeting production ("Behavioural patterns of vocal greeting production in four primate species").

2. "Greeting" calls are functionally defined calls. In this MS, authors term vocalizations, given during approach behavior of a focal animal "Greeting" calls. The function of these calls, however, is not yet known for the respective species and according to authors will not be addressed in this MS. Such a situation makes a comparison across species tricky since you may compare then apples with oranges.

We appreciate the Reviewer's concern. However, in our definition (in Table 1), "greeting" does not relate to the function of these calls, but to the situation in which they are produced. "Vocal greeting: vocal signal given during encounters (i.e., grunts for baboons and vervets, pant grunts for chimpanzees and grunts or twitters for sooty mangabeys)". We however now provide this definition in the text as well to avoid this confusion (L 68-70).

3. Abstract. Line 36: dominance rank. Do you mean dominance or rank order?

We now say "individual dominance rank" to avoid the confusion highlighted by the Reviewer (L 33).

4. In the Abstract, authors use the term "dyadic approach".

Please address in the methods the link between dyadic approach and an “encounter”.

We have reworded these parts of the abstract (L 33-43) and the Methods (L 169-170) to avoid the confusion highlighted by the Reviewer.

5. Introduction. Line 74. Please embed Scheumann et al. 2017 to refer to current knowledge on how primate “greeting” calls can be defined and studied (Scheumann, M.; Linn, S.; Zimmermann, E.: Vocal greeting during mother-infant reunions in a nocturnal primate, the gray mouse lemur (*Microcebus murinus*). In: Scientific Reports 7 (2017) 10321)

Thank you for this reference – we now cite this work (L 70 and 72).

6. Line 113: Please give reference for the divergence in the acoustic structure of “greeting” calls between the four studied species.

We have done this (L 121).

7. Methods. 157 ff and Discussion. As one of the former referees mentioned, not the same methods were applied for all species. This is still the case in the revised MS. Please outline in the discussion how this situation may have shaped your findings.

We appreciate the Reviewer’s comment and now discuss this particular aspect in the Discussion (L 471-477).

Further, one of the referees mentioned that authors did not present any data on the context of an encounter or the quality of an encounter. This issue remains in the revised MS and thus should be addressed in the discussion.

We now justify in the Discussion why “context” was not included in our analyses (L 514-523).

8. Line 258. Please outline the concept of Elo-rating here so that the general audience will be well informed upon the procedure you used here for comparative purposes. To refer to a reference is not sufficient to understand the content of your paper.

We have done this (L 263-274).

9. Abstract/Results: I still share the opinion of referee 2 that the following findings for mangabeys are difficult to understand: “low ranking animals were more likely to call than high-ranking ones” and “calling being negatively associated with rank distance ” (see e.g. Abstract). Please, address this issue in the discussion to make your findings better understandable for the general audience.

We have now amended this part of the Abstract to make it clear that dominance status is an individual feature whereas dominance distance is a dyadic feature (L 33, 36, 42 and 43).

10. Please check Figure 1, 2 and 5. In my version, I cannot see the description on the y-axis.

We are sorry for this inconvenience. The y-axis labels are correct - the line numbers are inserted automatically by the Manuscript Central when uploading the figures. The labels of the y-axis read as follows: Figure 1 – call proportion; Figure 2 and Figure 5– calling probability.

11. Please address whether all the primary data supporting the findings of the paper are publicly available.

We now make this clearer in the Data accessibility section (L 542).

12. Please give research permission no. in the paragraph Ethics.

We have done that for permits that were accompanied by a permit number.